# Antimicrobial Random Peptide Mixtures Eradicate *Acinetobacter baumannii* Biofilms and Inhibit Mouse Models of Infection

**DOI:** 10.3390/antibiotics11030413

**Published:** 2022-03-19

**Authors:** Hannah E. Caraway, Jonathan Z. Lau, Bar Maron, Myung Whan Oh, Yael Belo, Aya Brill, Einav Malach, Nahed Ismail, Zvi Hayouka, Gee W. Lau

**Affiliations:** 1Department of Pathobiology, University of Illinois at Urbana-Champaign, Urbana, IL 61802, USA; caraway3@illinois.edu (H.E.C.); jlau18@jhu.edu (J.Z.L.); oh31@illinois.edu (M.W.O.); 2Institute of Biochemistry, Food Science and Nutrition, The Robert H. Smith Faculty of Agriculture, Food and Environment, The Hebrew University of Jerusalem, Rehovot 76100, Israel; bmaron26@gmail.com (B.M.); yael.belo@mail.huji.ac.il (Y.B.); aya.brill@mail.huji.ac.il (A.B.); einav.cohen@mail.huji.ac.il (E.M.); 3Department of Pathology, College of Medicine, University of Illinois at Chicago, 840 South Wood Street, Chicago, IL 60612, USA; ismail7@uic.edu

**Keywords:** random peptide mixtures, *Acinetobacter baumannii*, antibiotic resistance, biofilms, acute pneumonia, soft tissue infection, sepsis

## Abstract

Antibiotic resistance is one of the greatest crises in human medicine. Increased incidents of antibiotic resistance are linked to clinical overuse and overreliance on antibiotics. Among the ESKAPE pathogens, *Acinetobacter baumannii*, especially carbapenem-resistant isolates, has emerged as a significant threat in the context of blood, urinary tract, lung, and wound infections. Therefore, new approaches that limit the emergence of antibiotic resistant *A. baumannii* are urgently needed. Recently, we have shown that random peptide mixtures (RPMs) are an attractive alternative class of drugs to antibiotics with strong safety and pharmacokinetic profiles. RPMs are antimicrobial peptide mixtures produced by incorporating two amino acids at each coupling step, rendering them extremely diverse but still defined in their overall composition, chain length, and stereochemistry. The extreme diversity of RPMs may prevent bacteria from evolving resistance rapidly. Here, we demonstrated that RPMs rapidly and efficiently kill different strains of *A. baumannii*, inhibit biofilm formation, and disrupt mature biofilms. Importantly, RPMs attenuated bacterial burden in mouse models of acute pneumonia and soft tissue infection and significantly reduced mouse mortality during sepsis. Collectively, our results demonstrate RPMs have the potential to be used as powerful therapeutics against antibiotic-resistant *A. baumannii*.

## 1. Introduction

The emergence of antimicrobial-resistant (AMR) microbial pathogens and the subsequent decrease in efficacy of antimicrobial drugs have severely exacerbated the threat of infectious diseases, affecting the ability of physicians to treat patients. The most recent Global Research on Antimicrobial Resistance (Gram) study estimates there were 4.95 million deaths associated with AMR in 2019, including 1.27 million deaths caused by bacterial AMR [1,2]. The rise of AMR is predominately linked to clinical overuse and overreliance on antibiotics, which have provided a strong selection pressure in bacterial populations that favors the development of resistant phenotypes [1,2,3]. Along with the overuse of antibiotics in clinical settings, the over-the-counter availability of these drugs, the increase in the use of antibiotics in livestock and poultry populations, and the under-regulation of waste disposal have also contributed to the substantial selection pressures present in contemporary bacteria populations [4,5,6]. The coupling of this strong selective pressure with the interconnected structure of modern human society has greatly contributed to the rapid rise of antibiotic-resistant bacteria strains across the globe. Another cause of the antibiotic resistance crisis is the lack of novel treatments against AMR bacteria due to severe financial and regulatory challenges faced by drug developers [7]. To combat this urgent issue, researchers must utilize a multifaceted approach which considers the challenges of novel drug development and avoids developing therapeutics that may cause severe selection pressure for the evolution of resistance seen with current conventional antibiotics.

Among the AMR bacterial species, the ESKAPE pathogens (*Enterococcus faecium*, *Staphylococcus aureus*, *Klebsiella pneumoniae*, *Acinetobacter baumannii*, *Pseudomonas aeruginosa*, and *Enterobacter* species) are a major cause of nosocomial (hospital-acquired) infections [8,9] and present a significant challenge due to dwindling treatment options. *A. baumannii* is an opportunistic pathogen that causes hospital-acquired and ventilator-associated pneumonia, bacteremia, meningitis, wound and surgical site infections (including necrotizing fasciitis), and urinary tract infections [10]. *A. baumannii* is considered an emerging global threat because various clinical isolates have become highly resistant, particularly to last-resort antibiotics such as carbapenems, a group of antibiotics widely prescribed for multidrug-resistant (MDR) infections [11,12]. Although there are multiple mechanisms underlying the carbapenem resistance, including mutations in the efflux pump [13,14] and outer membrane proteins [15], the primary driver is the acquisition of carbapenem-hydrolyzing oxacillinase-encoding genes [15,16], the expression of which is enhanced by the insertion of an upstream insertion element with a strong promoter [17,18,19]. Unsurprisingly, MDR clinical strains of *A. baumannii* are associated with increased chance of intensive care unit admittance, length of patient stays, ventilator use, and high morbidity and mortality [20,21]. 

The emergence of MDR, extensively drug resistant (XDR) and pandrug resistant (PDR) pathogens, caused by indiscriminate overuse of antibiotics, has driven the search for novel antimicrobial agents. An alternative approach to antibiotics is the use of antimicrobial peptides and proteins (AMPs) which naturally function in innate organismal immunity. AMPs are a large and diverse group of molecules deployed by multicellular organisms including humans and plants to combat pathogens and by microbes to leverage competitive advantages against competitors occupying the same niche [22,23,24]. AMPs and their synthetic derivatives display broad-spectrum antimicrobial activity. Their mechanisms of action are highly variable, and a particular AMP may employ more than one antibacterial mechanism, although a common property is their ability to bind and disrupt negatively charged bacterial membranes [25,26,27,28]. The activity of several synthetic AMPs has been recently demonstrated in mouse models of infection by both ESKAPE and other bacterial pathogens [29,30,31]. Although AMPs show promise, there are some major disadvantages. These include high cytotoxicity, low to moderate antimicrobial activity, low proteolytic stability, high cost of production [32,33,34], and the ability of bacteria to develop resistance [35,36]. Widespread application of AMPs has also caused the rise of bacterial strains with multiple resistance mechanisms [37,38]. For example, Perron et al. evolved and selected *Escherichia coli* and *Pseudomonas fluorescens* strains for resistance against pexiganan, an analogue of frog antimicrobial peptides (Magainins) [37]. 

Recently developed novel approaches may lead to more effective ways to fight antibiotic resistance pathogens. These include biogenic metallic nanoparticles (e.g., silver) with broad antibacterial and fungal activities [39,40]; adoption and refinement of pharmacophore approaches that couple empirical knowledge to molecular structural properties for rational antibiotic design and tailor patient therapy to tackle bacterial resistance [41,42,43]; and computational based simulation of model on choice of antibiotic usage and emergence of resistance to antibiotics [44]. 

Interestingly, a recent study has revealed that *S. aureus* evolved slower resistance to a mixture of two AMPs, suggesting that AMP cocktails have the potential to reduce the emergence of AMP resistance [45]. An attractive alternative to combinations of defined AMPs, recently developed by the Hayouka lab, is the use of ultra-diverse random peptides mixtures (RPMs) [46,47,48,49,50,51]. To generate these RPMs, we have modified the conventional solid-phase peptide synthesis methodology by adding a defined proportion of two amino acids at each coupling step, instead of one pure amino acid (Figure 1) [48]. After synthesizing the peptide mixture via n coupling steps, 2^n^ sequences of random peptides with a defined composition and controlled chain length are being generated. Each of the synthesized mixtures is random in terms of sequence but highly controlled in terms of chain length and stereochemistry. The RPM synthesis uses binary cationic-hydrophobic α-amino acid combinations. These RPMs have shown robust antimicrobial activity in in vitro bioassays towards “superbugs”, such as MRSA, vancomycin-resistant Enterococci (VRE), and *Listeria monocytogenes* [46,48,49,50,51]. Most recently, we have shown that these RPMs are non-cytotoxic toward cultured human bronchial epithelial cells [52]. Additionally, RPMs have strong safety profiles in mice, and importantly, exhibit high efficacy against *P. aeruginosa* and methicillin-resistant *S. aureus* (MRSA) in mouse models of acute pneumonia and sepsis [52]. In this study, we examined the efficacy of four different RPMs against *A. bauamannii*.

## 2. Results

### 2.1. Random Peptide Mixtures Are Effective Antibacterial Agents against A. baumannii In Vitro

We examined the antimicrobial activity of the following four RPMs as in our previous publication [52]—FK20 (L-Phenylalanine-L-Lysine, 20-mer), FdK (L-Phenylalanine-D-Lysine, 20-mer), dFdK (D-Phenylalanine-D-Lysine, 20-mer), and LK20 (L-Leucine-L-Lysine, 20-mer)—and against the MDR clinical *A. baumannii* strains W41979, F19521, and M13100, all of which are wound isolates from humans (Table 1). Overnight cultures of these *A. baumannii* strains were washed in sterile phosphate saline (PBS) and exposed to 4, 20, and 100 µg mL^−1^ of each peptide. Killing assays were performed in test tubes either on a rotating drum (e.g., planktonic growth) or under static incubation (e.g., biofilm formation conditions) to determine whether differences in metabolic rate (e.g., higher in rotating culture) confer resistance or susceptibility to RPMs. In rotating cultures, our pilot studies indicated that, at 4 µg mL^−1^, RPMs were ineffective against all three strains of *A. baumannii* (data not shown). At 20 µg mL^−1^, only dFdK and LK20 showed significant efficacy against strain W41979 (Figure 2a). Importantly, at 100 µg mL^−1^ all four RPMs were effective in killing strains W41979 and F19521, by a factor ranging from 2 to 4 log CFUs. Of the three strains, M13100 was most resistant to killing, with only FK20 showing 2.7 log killing at 100 µg mL^−1^. The killing of *A. baumannii* by RPMs (20 µg mL^−1^) did not appreciably change in the static cultures (Figure 2b), suggesting that metabolic differences may not be important in determining the susceptibility to RPMs within the short duration of one hour.

### 2.2. Random Peptide Mixtures in a Cocktail Are More Effective Than Single RPM and Show Additive Efficacy with Antibiotic in Killing A. baumannii In Vitro 

Because *A. baumannii* strain M13100 showed higher levels of resistance to RPMs, we examined whether a cocktail comprised of all four RPMs would provide better efficacy against M13100 and the other two strains *A. baumannii* in vitro [Figure 3]. In contrast to the inefficient killing by individual RPM, cocktail RPMs at the 20 µg mL^−1^ concentration successfully killed W41979, F19521, and M13100 by 1.5, 2.2, and 1.5 logs, respectively. The killing efficiency was amplified at the 100 µg mL^−1^ concentration and W41979 was killed at a factor of 3.2 logs, F19521 at 3.3 logs, and M13100 at 2.2 logs. More specifically, when compared against FdK, which was the only RPM that showed significant killing of M13100 at 20 µg mL^−1^, the killing was increased by a factor of 1.03 log. In addition, at the 100 µg mL^−1^ concentration, the cocktail RPMs showed comparable levels of killing to individual RPMs (Figure 3a). We further delineated which RPM killed strain M13100 by using various combinations of 2 RPMs (each at 10 µg mL^−1^). Interestingly, any combination of two peptides conferred significant killing of M13100, suggesting some synergistic or additive effect of these peptides against *A. baumannii* (Figure 3b). 

We further examined if RPMs could potentiate the killing of imipenem-resistant M13100. At 4, 10, and 20 µg mL^−1^, RPMs neither showed additive nor synergistic killing of M13100 with imipenem (Figure 3c, left panel). However, a combination of 100 µg mL^−1^ LK20 and 4 µg mL^−1^ imipenem killed M13100 at significantly higher level, demonstrating additive antibacterial effects between these two drugs (Figure 3c).

### 2.3. Random Peptide Mixtures Inhibit Biofilm Formation

Biofilms play a major role in a wide variety of infectious diseases, which the United States National Institute estimates are responsible for 80% of all infections [53]. Among others, biofilms associated diseases include urinary tract infections and vaginosis, otitis media, dental plaque, gingivitis, contact lenses, endocarditis, cystic fibrosis, and infections of permanent indwelling devices such as catheter, heart valves, and orthopedic implants [53,54,55,56,57,58]. Because the 100 µg mL^−1^ dose was most effective at killing *A. baumannii* in a planktonic bacterial culture, we subsequently tested if the same four RPMs at 100 µg mL^−1^ could also inhibit biofilm formation. Inhibition of static biofilm formation by RPMs was performed in *A. baumannii* strains cultured in 10% lysogeny Broth (LB) in test tubes and quantified with crystal violet staining as previously described [59]. *A. baumannii* treated with the same volume of sterile PBS served as control. We found that all four RPMs were similarly effective at inhibiting formation of biofilms when compared to PBS (Figure 4).

### 2.4. Random Peptide Mixtures Disrupt Preformed Biofilms

Microbial pathogens residing within biofilms are notoriously recalcitrant to antibiotics by multiple orders of magnitude [60,61]. Especially, persister cells are dormant variants and the main culprit within biofilm populations; they are highly tolerant to antibiotics and selected in chronic disease settings, including *P. aeruginosa* in cystic fibrosis and *Candida albicans* in oral candidiasis [60,61,62]. Therefore, we examine if RPMs could disrupt mature biofilms of *A. baumannii*. Static biofilms of F19521, M13100, and W41979 were allowed to form for 24 h before exposure to 100 µg mL^−1^ RPMs or sterile PBS control for another 24 h. As shown in Figure 5, all four RPMs effectively eradicated preformed biofilms. Collectively, the results in Figure 4 and Figure 5 demonstrate that RPMs have high potential in inhibiting both the process of biofilm formation as well as eradicating mature biofilms.

### 2.5. Random Peptide Mixtures Are Effective against Acute Pneumonia and Soft Tissue Infections in Mice

In our previous study concerning MRSA and *P. aeruginosa*, we have already shown that RPMs do not exhibit deleterious in vitro cytotoxicity towards human cells and demonstrate favorable safety profiles in mice [52]. We examined the pharmacodynamics of one of the RPMs, FK20, by tracking spatiotemporally the longevity and systemic spread of intravenously-injected (via retro-orbital route) FK20 conjugated to the fluorescence dye Cyanine7.5 (FK20-Cy7.5). As shown in Figure 6a, a large amount of FK20-Cy7.5 remained visible and continued to circulate systemically 24 h post injection. These observations suggest that FK20-Cy7.5 was rather resistant to proteolytic degradation mediated by host proteases, with steadied and prolonged presence after injection.

We have recently shown that RPMs could reduce the burden of *P. aeruginosa* and MRSA significantly in mouse models of acute pneumonia and sepsis [52]. As mentioned previously, *A. baumannii* is a major cause of hospital acquired pneumonia and ventilator-associated pneumonia [10]. We utilized acute pneumonia model to test the efficacy of intravenously administered RPMs at reducing the severity of *A. baumannii* infections in vivo. We chose to study *A. baumannii* strains W41979 and F19521 as they are more susceptible to killing by all four RPMs in vitro. Acute pneumonia infection was performed as we have previously published [52,63,64,65,66,67]. Infected mice were administered 5 mg/kg of dFdK, LK20, FK20, and FdK intravenously, twice daily. In the mice intranasally infected with *A. baumannii* strain W41979, we saw that dFdK, LK20, FK20, and FdK reduced the lung burden of infection by 2.78, 3.03, 1.96, and 1.35 log, respectively, when compared against PBS vehicle control (Figure 6b). Similarly, dFdK, LK20, FK20, and FDK reduced the lung burden of *A. baumannii* strain F19521 by 1.98, 1.63, 2.28, and 1.52 log, respectively. 

The neutropenic thigh soft tissue infection is the most frequently used model to assess the efficacy of new antibacterial agents because of minimal interference from host neutrophil-mediated bacterial clearance [68,69,70]. Mice were rendered neutropenic by injection of cyclophosphamide before thigh infection with *A. baumannii* strains W41979 and F19521. As shown in Figure 6c, dFdK, LK20, FK20, and FDK attenuated the thigh burden of W41979 by 3.00, 2.98, 2.61, and 1.98 log, respectively. Similarly, dFdK, LK20, FK20, and FdK attenuated the thigh burden of F19521 by 1.55, 1.83, 2.41, and 1.1 log, respectively. Collectively, these results show that all four RPMs have good pharmacokinetic properties, and they are effective at reducing bacterial burden in mouse models of acute pneumonia and soft tissue infection by *A. baumannii*.

### 2.6. Random Peptide Mixtures Attenuate Mortality in a Mouse Model of Sepsis

Because *A. baumannii* is a major cause of sepsis in humans, we evaluated whether the efficacy of RPMs could meet the stringent criterium of reducing mortality in a mouse model of sepsis caused by the highly pathogenic *A. baumannii* strain W41979 [67]. CD1 mice (cohorts of 15) were retro-orbitally infected with W41979 and treated twice daily with 10 mg/kg of intravenously administered dFdK or LK20. The control mouse cohort was treated with the same volume of sterile PBS. Infected mice treated with PBS exhibited 80% mortality rate within 100 h post-infection (Figure 7). In contrast, mice infected with W41979 and treated with dFdK or LK20 exhibited significantly higher survival rates and delayed kinetics in mortality (Figure 7), highlighting the therapeutic potential of RPMs in attenuating *A. baumannii* infections.

## 3. Discussion

Antibiotic resistance is one of the greatest challenges facing modern human medicine. Antimicrobial-resistant bacteria, including the ESKAPE pathogens, have increased the threat that many diseases pose to the world’s population and decreased the ability of physicians to treat patients quickly and effectively. Poor antibiotic stewardship, whether in clinical or food production settings, has exasperated this problem. Understanding and developing alternative treatments for antimicrobial-resistant pathogens is vital to fighting this emerging threat. One such possibility is utilizing AMPs, a diverse group of molecules that naturally function as part of the organismal innate immune system by disrupting bacterial membranes [22,23,24]. More specifically, the ultra-diverse synthetic random peptide mixtures, RPMs, are extremely promising as they maintain the potent membrane disrupting capabilities of AMPs and high efficacy against planktonic and biofilm-forming ESKAPE and other bacterial pathogens in vitro, including MRSA, VRE, *L. monocytogenes,* and *P. aeruginosa* [46,47,48,49,50,51,52,71,72]. RPMs are also effective in reduction of disease severity caused by several genera of plant pathogenic bacterial, including *Xanthomonas* [51], and inhibition of bacterial growth in dairy milk [71]. Additionally, RPMs have strong pharmacodynamic and safety profiles in mice, and importantly, exhibit strong efficacy by reducing bacterial burden *P. aeruginosa* and MRSA in mouse models of acute pneumonia and bacteremia [52].

*A. baumannii* is an opportunistic and emerging ESKAPE pathogen that can cause many forms of serious infection in humans, including pneumonia, bloodstream infections, meningitis, wound and surgical site infections (including necrotizing fasciitis), and urinary tract infections [10]. A previously published study estimated the annual global incidence of *A. baumannii* infection at >1,000,000 cases, of which, 50% are carbapenem-resistant cases [73], with a mortality rate reported from 8% to 35%, with the ventilator-associated pneumonia and bloodstream infections causing the highest mortality [74,75]. In this study, we establish that multiple RPMs can effectively kill multiple strains of *A. baumannii* in vitro, inhibit biofilm formation, and disrupt preformed biofilms in vitro. RPMs show promising efficacy by reducing bacterial burden against acute pneumonia and neutropenic soft tissue thigh infection by two MDR clinical strains of *A. baumannii* W41979 [67] and F19521. Most significantly, RPMs dFdK and LK20 reduce mortality in preclinical mouse model of sepsis by *A. baumannii* W41979. Another important finding is that the RPM LK20 showed additive efficacy with imipenem against a resistant *A. baumannii* strain. When coupled with our recent report showing that RPMs were not cytotoxic in vitro and did not exhibit abnormal levels of toxicity in preclinical mouse models [52], the current study demonstrates that RPMs are attractive alternatives to ever dwindling choices of antibiotics to combat infections caused by carbapenem-resistant *A. baumannii* (Table 1). Future efforts will examine the efficacy of RPMs against other strains of MDR *A. baumannii* in additional models of human infectious diseases by *A. baumannii*, as well as investigate in depth the synergism between RPMs and carbapenems/other antibiotics. 

Our results show the ability of RPMs at inhibiting biofilm formation as well as disrupting preformed *A. baumannii* biofilms. It has been estimated that 40–80% of bacterial species have the capability to form biofilms [76]. Pathogenic bacteria (e.g., *L. monocytogenes*) forming biofilms inside food processing facilities could lead to spoilage, endangering consumer health [77]. In healthcare settings, biofilms can persist on medical devices and on patients’ tissues [53,54,55,56,57,58] and are estimated to be responsible for 80% of all nosocomial infections [53]. Bacterial biofilms protect and create a more favorable niche against harsh environmental insults, and their disruption is key for halting bacterial growth. Biofilms shield bacteria from the host’s immune system, leading to chronic and persistent infections. Furthermore, biofilm bacterial infections can lead to local and collateral tissue damage and have been shown to be more resistant to antibiotics [78]. The strong and consistent disruption of these biofilms further illustrates the promise of RPMs as a new class of antimicrobials. 

RPMs contain complex mixtures of slightly different peptides, and the observed antimicrobial activity is the average of the whole population. As further support to this proposed approach, we found that the combination of two or four RPMs was able to synergistically increase the killing of all three *A. baumannii* strains at 20 μg mL^−1^ concentration significantly, in contrast to lack of killing by individual RPM given at the same concentration. Similar antimicrobial synergism was previously observed in short lipo-RPMs (derived from N-palmitoylation of RPMs) [72]. Furthermore, we predict that complex heterogeneous mixtures of RPMs that display strong and broad antimicrobial activity will challenge bacteria to sense and respond to a perceived threat and significantly delay or even abolish the evolution of resistance. RPMs can be readily synthesized via a solid phase peptide synthesis methodology, which enables access to a wide variety of cationic random peptide mixtures. One potential challenge with the RPMs approach lies in the batch-to-batch biosimilarity of each RPM. However, this does not appear to be a problem and can be addressed by comparing several batches of each RPM for chemical characterization and antimicrobial activity as our results were derived from multiple batches of RPMs. Similar consistency was recently demonstrated in lipo-RPMs by mass spectrometry [72].

Further studies are needed to explore the efficacy of RPMs against other bacterial pathogens in animal models of infectious diseases. The combined results of this study and those previously found by Bennet et al. [52] illustrate that RPMs are broad spectrum antibacterials that effectively target multiple bacterial species. Future efforts will be targeted on using RPMs against *Candida auris*, *Clostridioides difficile*, and carbapenem-resistant enterobacteriaciae (CRE) such as *Klebsiella pneumoniae* and *Enterobacter cloacae*. The effectiveness of RPMs against the MDR ESKAPE remains to be seen but should be addressed to demonstrate the scope of uses for these potential antibiotic replacements. Additional effort will be devoted to examining mechanisms underlying the synergy between RPMs (e.g., LK20) and antibiotics.

## 4. Materials and Methods

### 4.1. Synthesis and Storage of the Random Peptide Mixtures

RPMs were synthesized as we have previously described [46,49,50,51]. The success of the synthesis was confirmed by MALDI-TOF mass spectrometry and amino acid analyses that validated the molecular weight and the ratio between the two amino acids that were used to synthesize the RPMs [47,72]. The RPMs were diluted to desirable concentrations in sterile PBS and stored at −20 °C. Freshly thawed RPMs were used for each experiment to avoid loss of activity caused by repeated freeze/thaw cycles.

### 4.2. Bacterial Cultures and MIC Determination

#### 4.2.1. Bacterial Culture Conditions

*A. baumannii* strains W1947, F19521, and M13100 were clinical isolates obtained from the University of Illinois at Chicago School of Medicine by our coauthor Dr. Ismail, as we have previously published [67]. *A. baumannii* strains were cultured in LB (Benton, Dickinson) at 37 °C overnight (~12 h). Bacteria were then resuspended in a 20% sterile glycerol/80% LB mixture and maintained at −80 °C. Prior to experimentation, bacteria were subcultured from the frozen stocks into fresh liquid LB (OD600 nm ~0.01) and grew overnight to stationary phase (OD600 nm ~3.0) as determined by using a spectrophotometer. Bacteria were washed in sterile PBS and diluted to appropriate concentrations for individual assays. The number of viable bacteria (CFU) was determined by plating onto LB agar plates after serial dilution.

#### 4.2.2. Measurement of the Anti-Microbial Susceptibility Profile of Bacterial Isolates

The antimicrobial susceptibility phenotype of *A. baumannii* strains were determined using the MicroScan Walkaway, a broth microdilution, antimicrobial susceptibility test system, and Gram-negative panels by experienced medical technicians as per the manufacturer’s instructions. The clinical isolates strains were first subcultured on blood agar (18 to 24 h, 35  ±  2 °C). Various antimicrobial agents were diluted in Mueller–Hinton broth supplemented with calcium and magnesium to concentrations bridging the range of clinical interest. The bacterial suspension (0.5 McFarland standard) was prepared in saline by direct colony suspension method using a nephelometer (Thermo Fisher Scientific). After inoculation of the Gram-negative panel and rehydration with a standardized suspension of organism and incubation at 35 °C for a minimum of 16 h, the MIC for *A. baumannii* isolates were determined by observing the lowest antimicrobial concentration showing inhibition of growth. Panels containing ceftazidime, ceftriaxone, aztreonam, or cefotaxime at 1 µg/mL or cefpodoxime at 1 or 4 µg/mL (depending on panel type) can be used to screen for Gram-negative bacteria suspected of producing extended-spectrum beta-lactamases (ESBLs). Panels containing ceftazidime/clavulanic acid and cefotaxime/clavulanic acid can be used to confirm the presence of ESBLs. The confirmation test is a ≥3 two-fold dilution decrease in MICs of suspected organisms to ceftazidime or cefotaxime in the presence of a fixed concentration of clavulanic acid, versus its MIC when tested alone. Interpretation of MICs for each drug against *Acinetobacter* as susceptible, intermediate, and resistant were determined according to the Clinical and Laboratory Standard Institute (CLSI) M100 (performance standards for antimicrobial susceptibility testing) guidelines.

### 4.3. In Vitro A. baumannii Killing Assays

Cultures of *A. baumannii* were grown overnight at 37 °C. After centrifugation, bacterial pellets were washed three consecutive times with sterile PBS. Washed bacteria were serially diluted to 107–108 CFU mL^−1^ in PBS and supplemented with 4 µg mL^−1^, 20 µg mL^−1^, or 100 µg mL^−1^ of the RPMs. For the control, an equal volume of PBS was used. The *A. baumannii* and RPM mixtures were rotated on a TC-7 rotary drum (New Brunswick Scientific, Enfield, CT, USA) at speed setting six and maintained at 37 °C for 60 min. For static bacterial killing, *A. baumannii* strains were incubated statically without rotation at 37 °C for 60 min. Samples were transferred and chilled on ice, followed by 1:10 serial dilutions using chilled PBS. Samples were immediately plated on LB agar and incubated overnight at 37 °C and CFU enumerated. Bacterial killing assays were performed in triplicate and independently on three separate occasions. 

### 4.4. Biofilm Inhibition and Disruption Assays

The biofilm inhibition assays were performed according to a previously published protocol [59]. *A. baumannii* strains W41979, F19521, and M13100 were cultured in LB overnight (OD600 nm ~3.0). Then, 10 µL of the culture was added to a test tube containing 900 µL of 10% LB and 100 µg of the RPM, with 100 µL of PBS as the control. The glass tubes were incubated statically at 37 °C 24 h, after which the non-adhering bacterial supernatant was discarded. The tubes were gently rinsed with water three times and allowed to air dry. Then, 1 mL of a 0.1% CV solution was added to stain any adhered biofilms at room temperature for 30 min. The dye was discarded, and the tubes were gently rinsed three times with water again. After air drying, 1 mL of 70% ethanol was added to each tube to dissolve the dye by gentle vortexing, and the saturation of the liquid was photographed. The dissolved crystal violet dye solution was transferred to cuvettes, and the number of biofilms was then quantified by measuring absorbance at 595 nm with a spectrophotometer. Biofilm disruption assays were performed in the same manner, except that the biofilms were allowed to form for 24 h before the addition of RPMs in fresh 10% LB. All biofilm inhibition and disruption assays were performed in triplicate and independently three times.

### 4.5. Mouse In Vivo Imaging and Infection Studies

Mouse studies were performed in strict accordance with the Guide for the Care and Use of Laboratory Animals of the National Institutes of Health. Animal protocols were vetted and approved by the Institutional Animal Care and Use Committee (IACUC) at the University of Illinois at Urbana-Champaign. CD-1 mice (6–7 weeks old, both males and females) were purchased from Charles River, and mice were acclimated for one week before experimentation. All mice were housed in positively ventilated microisolator cages with automatic recirculating water located in a room with laminar, high efficiency particulate-filtered air. The animals received autoclaved food, water, and bedding.

#### 4.5.1. In Vivo Imaging of FK20-Cy7.5 Pharmacodynamics

For the in vivo imaging, CD-1 mice (n = 2, male, 7-weeks old) were anesthetized with 3% isoflurane in an induction chamber. Mice were intravenously injected with FK20-Cy7.5 (50 μg) through the retro-orbital route and imaged using an IVIS SpectrumCT imaging system (PerkinElmer). Fluorescence images were acquired using the following settings: binning factor as 1, f number as 1, field of view as 25.4, and fluorescence exposure time for 60 s. Images were analyzed by Living Image^®^ Software (PerkinElmer, Waltham, MA, USA). 

#### 4.5.2. Mouse Model of Acute Pneumonia

CD-1 mice (males and females, in cohorts of 7–8) were intranasally inoculated with *A. baumannii* strains W19547 or F19521 (please refer to legend of Figure 6 for inoculum concentrations). Infected mice were intravenously treated twice daily for two days with 5 mg/kg dFdK, LK20, FK20, or FdK. Control cohorts were treated with same volume of sterile PBS. At 48 h, mice were euthanized, and the lungs harvested and homogenized with an Omni Soft Tissue Tip™ Homogenizer (OMNI International, Tulsa, OK, USA) in 1 mL of sterile PBS. Bacterial burden in the tissue homogenates was determined by serial dilution plating onto LB agar.

#### 4.5.3. Mouse Model of Neutropenic Soft Tissue Thigh Infection

CD-1 mice (males and females, cohorts of 7–8) were rendered neutropenic by intraperitoneal injection of cyclophosphamide (150 mg/kg on Day-5, and 100 mg/kg Day-2). On Day 1, mice were anesthetized with xylazine/ketamine, and fur on the right hind thigh was removed by clipping with a pair of scissors followed by application of depilating gel. After 24 h, mice were anesthetized with isoflurane and infected with *A. baumannii* strains W19547 or F19521 (please refer to legend of Figure 6 for inoculum concentrations) by injection into the thigh muscle (bicep femoris) with a 30 G needle. Infected mice were intravenously treated twice daily for two days with 5 mg/kg dFdK, LK20, FK20, or FdK. Control cohorts were treated with same volume of sterile PBS. Infected animals were monitored for myositis and lameness until euthanasia. At 48 h, mice were euthanized, and infected thigh muscle tissues were harvested and homogenized with an Omni Soft Tissue Tip™ Homogenizer (OMNI International) in 2 mL of sterile PBS. Bacterial burden in the tissue homogenates was determined by serial dilution plating onto LB agar.

#### 4.5.4. Mouse Model of Bacterial Sepsis Survival

Infection was established via 100 µL retro-orbital injection of *A. baumannii* strain W41979 (please refer to legend of Figure 7 for inoculum concentrations). Mice were treated twice daily intravenously for three days with 10 mg/kg dFdK or LK20. Control cohorts were treated with same volume of sterile PBS. Mouse mortality was monitored for 100 h.

### 4.6. Statistical Analysis

Quantitative data was expressed as the mean ± standard deviation and was calculated using their respective functions. Statistical significance was determined by using the Student’s *t*-test. We used an unpaired two-tailed Student’s t-test for the analysis of two groups. Statistical significance was expressed as *p* ≤ 0.05, *p* ≤ 0.01, *p* ≤ 0.001, *p* ≤ 0.001, or ns (not significant). For survival analyses, a Kaplan–Meier Log Rank Survival Test was performed using the GraphPad Prism Version 9.0.2 Software.

## Figures and Tables

**Figure 1 antibiotics-11-00413-f001:**
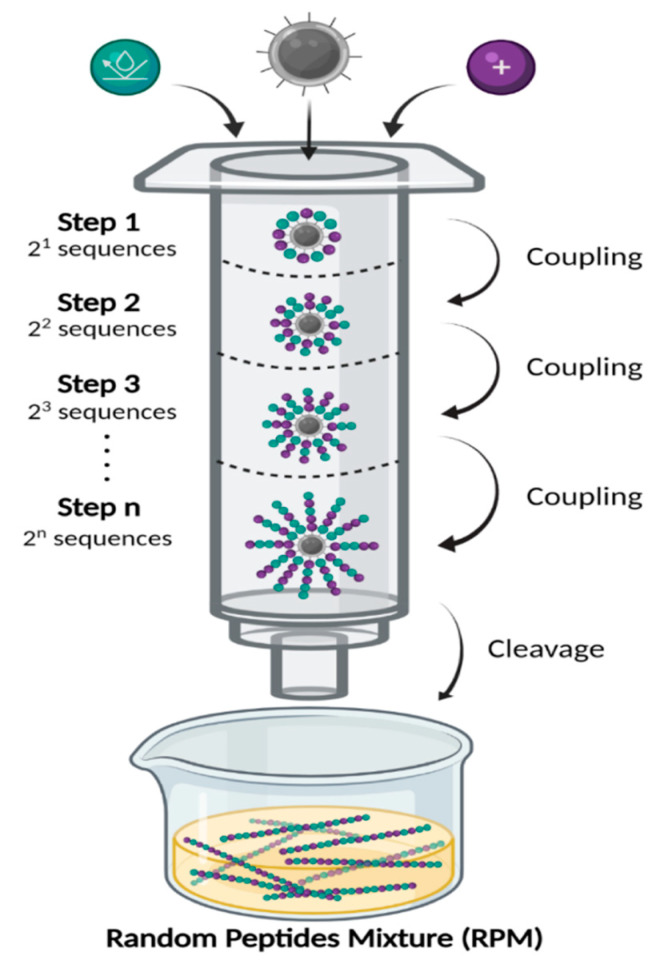
Random peptide mixtures (RPMs) synthesis. RPMs were synthesized using standard Fmoc-based solid-phase peptide synthesis (SPPS), on Rink amide resin (gray). Before each coupling step, the Fmoc protecting group was removed, then fresh solution containing 1:1 ratio of cationic (purple) and hydrophobic (cyan) protected amino acids was added. In each step there are 2^n^ optional peptide sequences (n = number of coupling steps). After the desired number of coupling steps, the peptides were cleaved from the solid support to generate free randomized linear peptides.

**Figure 2 antibiotics-11-00413-f002:**
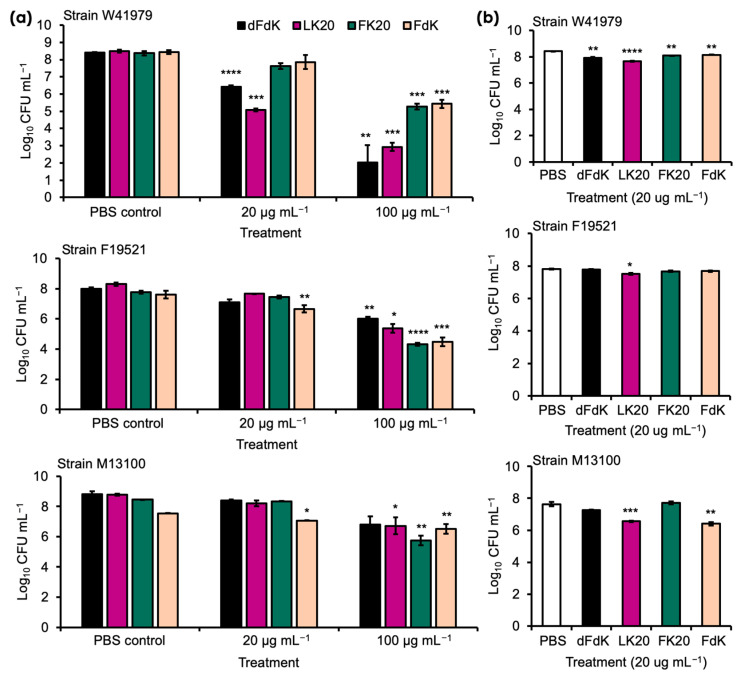
Random peptide mixtures (RPMs) effectively kill *A. baumannii* in vitro. Overnight stationary phase cultures of *A. baumannii* strains W41979, F19521, and M13100 were diluted to 10^7^–10^8^ CFU mL^−1^ before exposure to indicated concentrations of RPMs for 1 h by incubating at 37 °C rotating on a rotary drum (**a**) or under static conditions (**b**). The extent of bacterial killing was determined by colony forming unit (CFU) after serial dilution. Error bars represent SD for n = 3 of a typical experiment performed independently three times. Significance was determined with Student’s *t*-test. * *p* < 0.05, ** *p* < 0.01, *** *p* < 0.001, and **** *p* < 0.0001 significance cmpared to respective phosphate buffered saline (PBS) control; data without symbols are not statistically significant.

**Figure 3 antibiotics-11-00413-f003:**
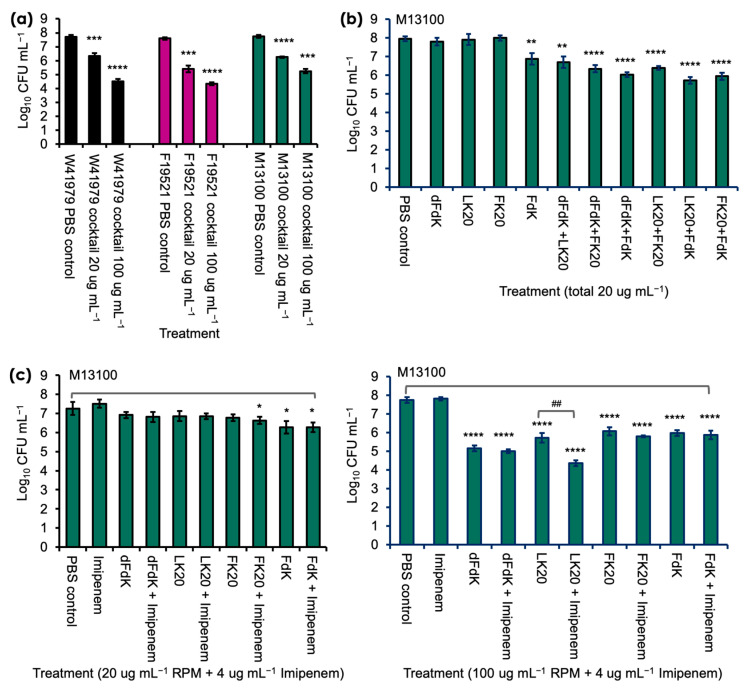
Random peptide mixtures (RPMs) cocktail kills *A. baumannii* more effectively than individual RPMs and shows additive effects with imipenem. (**a**) The 20 μg mL^−1^ cocktail was comprised of 5 μg mL^−1^ each of dFDK, LK20, FK20, and FdK whereas the 100 μg mL^−1^ cocktail was comprised of 25 μg mL^−1^ each of the aforementioned RPM. (**b**) Killing of *A. baumannii* strain M13100 by individual or cocktail of two RPMs (each RPM at 10 μg mL^−1^). (**c**) Killing of M13100 by RPM and imipenem cocktails. Error bars represent SD for n = 3 of a typical experiment performed independently three times. * *p* < 0.05, ** *p* < 0.01, *** *p* < 0.001, and **** *p* < 0.0001 significance compared to respective phosphate buffered saline (PBS) control as determined by the Student’s *t*-test. ## *p* < 0.01 significance compared to respective LK20 against LK20 + imipenem as determined by the Student’s *t*-test.

**Figure 4 antibiotics-11-00413-f004:**
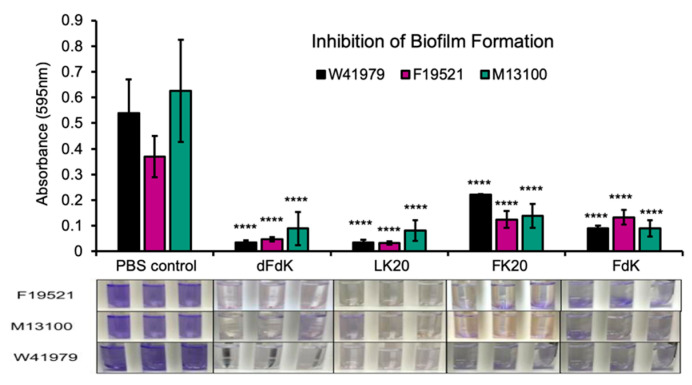
Inhibition of *A. baumannii* biofilm formation by random peptide mixtures (RPMs) in vitro. *A. baumannii* strains F19521, M13100, and W41979 were cultured in 10% lysogeny broth (LB) under static condition for 24 h in the presence of 100 µg mL^−1^ RPMs or sterile phosphate buffered saline (PBS). The number of biofilms was determined by crystal violet staining. Error bars represent SD for n = 3 of a typical experiment performed independently three times. Significance was determined with the Student’s *t*-test. **** *p* < 0.001 significance compared to respective PBS control.

**Figure 5 antibiotics-11-00413-f005:**
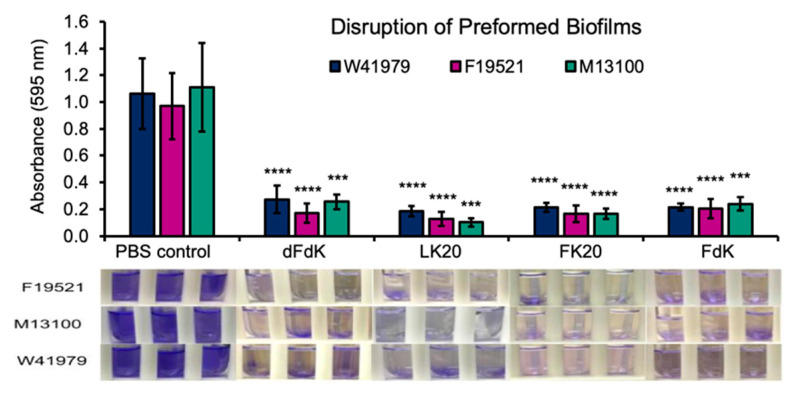
RPMs disrupt preformed *A. baumannii* biofilms. *A. baumannii* strains F19521, M13100, and W41979 were allowed to form biofilms statically in 10% lysogeny broth (LB) for 24 h before exposure to 100 µg mL^−1^ random peptide mixtures (RPMs) or phosphate buffered saline (PBS) for another 24 h. The number of biofilms was determined by crystal violet staining. *** *p* < 0.001 and **** *p* < 0.0001 significance compared to respective controls as determined by the Student’s *t*-test.

**Figure 6 antibiotics-11-00413-f006:**
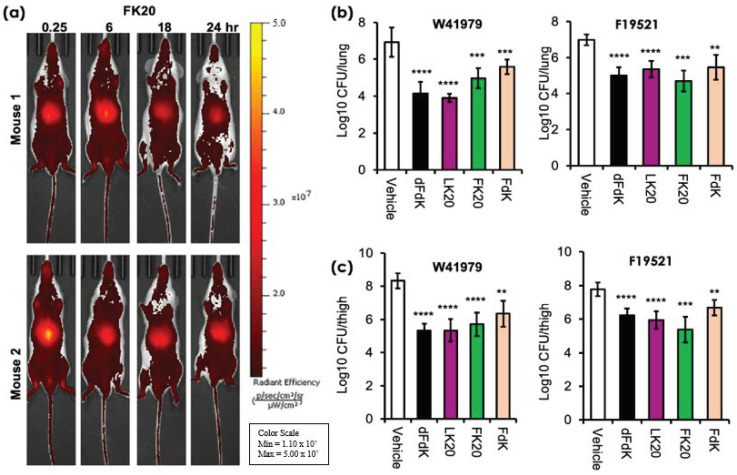
In vivo pharmacodynamics and efficacy of random peptide mixtures (RPMs). (**a**) Pharmacokinetics of intravenously injected Cyanine7.5-labeled FK20 (50 µg) in CD-1 mice (n = 2) and imaged for 24 h using an IVIS SpectrumCT imaging system. (**b**,**c**) Effectiveness of RPMs against *A. baumannii* infection. For acute pneumonia, CD-1 mice (n = 6–8) were intranasally-challenged with *A. baumannii* strain W41979 (1.4 × 10^8^ CFU/mouse) or F19521 (9.2 × 10^7^ CFU/mouse). For soft tissue infection, CD1 mice were rendered neutropenic by cyclophosphamide and infected with W41979 (1.1 × 10^5^ CFU/mouse) or F19521 (2.7 × 10^5^ CFU/mouse) intramuscularly. Infected mice were treated with either 5 mg/kg of each RPM or sterile phosphate buffered saline (vehicle) twice daily, for 2 days before determination of bacterial burden. ** *p* < 0.01, *** *p* < 0.001, and **** *p* < 0.0001 significance compared to respective vehicle controls as determined by the Student’s *t*-test.

**Figure 7 antibiotics-11-00413-f007:**
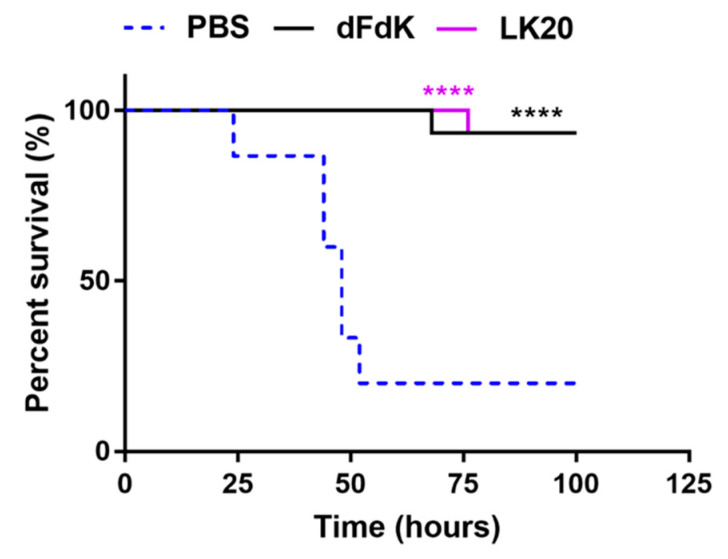
Random peptide mixtures (RPMs) attenuate mouse mortality during sepsis infection. CD1 mice (cohorts of 15) were retro-orbitally-infected with *A. baumannii* strain W41979 (9.65 × 10^7^ CFU). Infected mice were treated with dFdK or LK20 (10 mg/kg) or sterile phosphate buffered saline (PBS) twice daily, for 3 days. Mouse survival was monitored for 100 h. **** *p* < 0.0001 were derived when comparing the mortality of infected mice treated with RPMs against those treated with PBS by using the Kaplan–Meier Log Rank (Mantel–Cox) survival test.

**Table 1 antibiotics-11-00413-t001:** Antibiotic susceptibility testing of *A. baumannii* strains.

	Antibiotic	MIC (μg mL^−1^)	Interpretation ^1^
**Strain W41979**	Ceftazidime	>64	R
	Cefepime	32	R
	Gentamycin	>16	R
	Levofloxacin	4	I
	Pipperacillin/Tazobactam	>128	R
	Ciprofloxacin	>2	R
	Tobramycin	>16	R
	Imipenem	>16	R
	Meropenem	>8	R
	Colistin	0.19	S
**Strain F19521**	Ceftazidime	>64	R
	Cefepime	>64	R
	Gentamycin	>16	R
	Levofloxacin	4	I
	Pipperacillin/Tazobactam	>128	R
	Ciprofloxacin	>2	R
	Tobramycin	>16	R
	Imipenem	>16	R
	Meropenem	>8	R
	Colistin	0.125	S
**Strain M13100**	Amikacin	4	S
	Cefazolin	>32	R
	Ceftazidime	>64	R
	Cefepime	>64	R
	Ceftriaxone	>64	R
	Gentamycin	>16	R
	Levofloxacin	4	I
	Pipperacillin/Tazobactam	>128	R
	Trimethoprim/sulphamethox	>320	R
	Ampicillin/sulbactam	>32/16	R
	Tobramycin	>16	R
	Imipenem	4	R
	Erapenem	8	R

^1^ S = Sensitive; R = Resistant; I = Intermediate.

## Data Availability

Not applicable.

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
