# Peer review of "Antimicrobial Random Peptide Mixtures Eradicate Acinetobacter baumannii Biofilms and Inhibit Mouse Models of Infection"

_antibiotics, 2022, doi:10.3390/antibiotics11030413_

Round 1

Reviewer 1 Report

In this study, the authors used cocktail of random peptide mixture (RPM) and showed that these RPMs rapidly and efficiently kill different strains of A. baumannii, inhibit biofilms formation, and disrupt mature biofilms. Further authors also reported that use of the RPMs attenuated bacterial burden in mouse models of acute pneumonia and soft tissue infection, moreover, use of these RPMs significantly reduced mouse mortality during sepsis. Overall, the data is presented clearly and are analyzed in an appropriate manner. However, some of the conclusions made by the authors could be supported by additional experimental controls.

Major comments

  • Authors reported that combination of 4 RPMs was able to synergistically increase the killing. It could be possible that only 2 RPMs in the mixture are required to observe effective killing, while others are redundant. To test this authors can do killing assay using different combinations of 4 RPMs and find out which combination is more effective than a single RPM.
  • Authors reported that effective killing of M13100 was observed only with FK20 at 100 µg/mL, while M13100 biofilm formation was inhibited will all the 4 RPMs used in this study. This difference in the phenotype can be attributed the way experiments were done. Biofilms experiments were performed in static cultures, which generally create microaerophilic conditions, while killing assays were done in rotatory drum. Since bacteria are metabolically more active when shaking/rotating in comparison to static, this will make lot of difference in resistance towards RPMs. I would suggest authors to repeat the killing assay exactly same conditions how they performed biofilms.
  • Does usage of RPMs together with antibiotics will increase the killing? Authors can test this by using MICs of antibiotic in combination with RPMs to check whether this will have any additive/synergetic effect on bacteria.
  • How are quantified biofilms normalized? Since RPMs are killing the cells, did authors determined the CFUs of supernatants before quantifying the biofilms? If cells are dying when treated with RPMs then there will be definitely reduced biofilm formation.

Minor comments

  • Line 116 activity spelling need to corrected
  • Line 124 should be Figure 2
  • Table-1 : Did not mention any where how MICs are determined . If this data is not from this study, reference need to be added.
  • Line 361-369 Require more details:  Staring OD 600?  OD of the culture when cells were harvested for treatment with RPMs?
  • Figures : Colors of the figures need to be consistent for the strains and RPMs so that it will easy to follow.

Author Response

We would like to thank Reviewer 1 all the constructive comments to further improve our manuscript entitled “Antimicrobial Random Peptide Mixtures Eradicate Acinetobacter baumannii Biofilms and Inhibit Mouse Models of Infection” (antibiotics-1615871). Below, we have tried our best to address various queries. Reviewer’s critiques are in bold.

Reviewer 1

Major comments

  1. Authors reported that combination of 4 RPMs was able to synergistically increase the killing. It could be possible that only 2 RPMs in the mixture are required to observe effective killing, while others are redundant. To test this authors can do killing assay using different combinations of 4 RPMs and find out which combination is more effective than a single RPM.

Response: We thank Reviewer 1 for this suggestion. We have analyzed the various combinations of 2 RPMs, and indeed, found them more efficacious than a single RPM. The new data are included in Figure 3b.

  1. Authors reported that effective killing of M13100 was observed only with FK20 at 100 µg/mL, while M13100 biofilm formation was inhibited will all the 4 RPMs used in this study. This difference in the phenotype can be attributed the way experiments were done. Biofilms experiments were performed in static cultures, which generally create microaerophilic conditions, while killing assays were done in rotatory drum. Since bacteria are metabolically more active when shaking/rotating in comparison to static, this will make lot of difference in resistance towards RPMs. I would suggest authors to repeat the killing assay exactly same conditions how they performed biofilms.

Response: We have performed the static killing assays of all three A. baumannii strains with 20 mg/ml of RPMs. The killing rates are similar to rotating cultures at 20 mg/ml of RPMs. These new results are included as Figure 2B. Thus, the most likely explanation for strong inhibition of biofilm formation and disruption of preformed biofilms of M13100 is the different incubation time with the peptide between treatments (1 hour in rotating killing assays versus 24 hours in biofilm inhibition and disruption assays).

  1. Does usage of RPMs together with antibiotics will increase the killing? Authors can test this by using MICs of antibiotic in combination with RPMs to check whether this will have any additive/synergetic effect on bacteria.

Response: We have used the A. baumannii strain M13100, which is the most resistant to RPMs among the three strains we have studied, to examine the synergy/additive effects between Imipenem (4 mg/ml MIC) and RPMs (4, 10, 20 and 100 mg/ml).  Significant additive effect was only observed between Imipenem (4 mg/ml MIC) and the RPM LK20 at 100 mg/ml concentration (see Figure 3C).

  1. How are quantified biofilms normalized? Since RPMs are killing the cells, did authors determined the CFUs of supernatants before quantifying the biofilms? If cells are dying when treated with RPMs then there will be definitely reduced biofilm formation.

Response: For the biofilm inhibition assays, the overnight cultures were adjusted to the same OD600 nm, and yielded similar CFUs on agar plates. We agree with the reviewer that since RPMs are killing the A. baumannii, biofilms inhibition assays are not as informative. For that reason, we have also analyzed and demonstrated that RPMs disrupt performed biofilms.

Minor comments

  1. Line 116 activity spelling need to corrected

Response: Corrected

  1. Line 124 should be Figure 2.

Response: Corrected

  1. Table-1 : Did not mention anywhere how MICs are determined . If this data is not from this study, reference need to be added.

Response: MIC testing information is now included in the Materials and Methods section.

  1. Line 361-369 Require more details:  Starting OD 600?  OD of the culture when cells were harvested for treatment with RPMs?

Response: Prior to experimentation, bacteria were subcultured from the -80o C frozen stocks into fresh liquid LB (OD 600 nm ~0.01) and grew overnight to stationary phase (OD 600 nm ~3.0) as determined by using a spectrophotometer. Bacteria were washed in sterile PBS and diluted to appropriate concentrations for individual assays.

  1. Figures : Colors of the figures need to be consistent for the strains and RPMs so that it will easy to follow

Response: As requested, we have changed the color labeling for both A. baumannii strains and RPMs for consistency.

Reviewer 2 Report

The manuscript entitled "Antimicrobial Random Peptide Mixtures Eradicate Acinetobacter baumannii Biofilms and Inhibit Mouse Models of Infection" presents the results of a study devoted to the study of the antimicrobial effect of mixtures of short synthesized peptides of random amino acid composition. This topic is in demand and of interest in the context of the progressive development of resistance of pathogenic bacteria to widely used antibiotics. The authors studied one of the most relevant bacteria included in ESKAPE, which makes their work particularly significant. The article is written in understandable language, the research is carried out correctly, the results are calculated reliably. The article can be accepted after minor changes.

There are several questions to the authors and recommendations for correcting the text of the article.

1) Pages 6-7. Figure 4 and its signature should preferably be placed on the same page.

2) Line 349. It is advisable to move the link to the site to References, to facilitate perception.

3) It is desirable to place the Materials and Methods chapter before the Results chapter. This will eliminate the need to make cumbersome captions to the figures containing a description of the methods.

4) Lines 406, 413, 421, 435. At the beginning of the lines, in the number, there is an extra dot.

5) Line 629. The link to the article omitted bibliographic information. 56(80), 12053-12056.

6) Lines 132-139. The description of the methodology should be moved to the Materials and Methods chapter. In the Figure caption, it is desirable to add a transcript of all abbreviations and brief information necessary to understand the figure.

7) Lines 154-159. The description of the methodology should be moved to the Materials and Methods chapter. In the Figure caption, it is desirable to add a transcript of all abbreviations and brief information necessary to understand the figure.

8) Lines 175-184. The description of the methodology should be moved to the Materials and Methods chapter. In the Figure caption, it is desirable to add a transcript of all abbreviations and brief information necessary to understand the figure.

9) Lines 197-203. The description of the methodology should be moved to the Materials and Methods chapter. In the Figure caption, it is desirable to add a transcript of all abbreviations and brief information necessary to understand the figure.

10) Lines 216-230. The description of the methodology should be moved to the Materials and Methods chapter. In the Figure caption, it is desirable to add a transcript of all abbreviations and brief information necessary to understand the figure.

11) Lines 267-272. The description of the methodology should be moved to the Materials and Methods chapter. In the Figure caption, it is desirable to add a transcript of all abbreviations and brief information necessary to understand the figure.

12) Lines 206-208. RPM, according to your data, are not cytotoxic, which contradicts the researchers' data, which is written at the beginning (lines 82-84). There is a need to explain what the differences in the results are related to. If your data are given for specific RPMs, it would be desirable to explain how they differ favorably from those RPMs that are described as cytotoxic.

13) In general, given the random nature of the assembly of peptides, there are two legitimate questions. Are the results reproducible, given the large number of random synthesis options? How possible is it that when analyzing the cytotoxicity of a mixture of peptides, a toxic peptide will be missed, which may be accidentally synthesized later?

14) Is there a potential threat that foreign peptides that have entered the body will form into dangerous structures like prions?

15) It is desirable to supplement the Introduction chapter with new data on research and development in the search for new methods to counteract bacterial resistance to antibiotics, for example, https://doi.org/10.3390/mi12121480 , https://doi.org/10.1080/17460441.2022.1985459 , https://doi.org/10.1186/s13662-021-03423-8.

Author Response

We would like to thank Reviewer 2 for all the constructive comments to further improve our manuscript entitled “Antimicrobial Random Peptide Mixtures Eradicate Acinetobacter baumannii Biofilms and Inhibit Mouse Models of Infection” (antibiotics-1615871). Below, we have tried our best to address various queries. Reviewer’s critiques are in bold.

Reviewer 2

  1. Pages 6-7. Figure 4 and its signature should preferably be placed on the same page.

Response: We have moved Figure 4 and its legend to the same page.

  1. Line 349. It is advisable to move the link to the site to References, to facilitate perception.

Response: The link no longer exists on the US CDC website. Thus, we have removed it from the main text.

  1. It is desirable to place the Materials and Methods chapter before the Results chapter. This will eliminate the need to make cumbersome captions to the figures containing a description of the methods.

Response: The sequence where Materials and Methods appear in the manuscript is dictated by the journal (e.g., Title Page, Abstract, Introduction, Results, Discussion, Materials and Methods). Wherever appropriate, we have shortened the methodology description in all figures.

  1. Lines 406, 413, 421, 435. At the beginning of the lines, in the number, there is an extra dot.

Response: We thank Reviewer 2 for his astute observations. We have corrected these errors.

  1. Line 629. The link to the article omitted bibliographic information. 56(80), 12053-12056.

Response: We apologize for the error. It appears that the journal staff may have fixed the problem.

  1. Lines 132-139. The description of the methodology should be moved to the Materials and Methods chapter. In the Figure caption, it is desirable to add a transcript of all abbreviations and brief information necessary to understand the figure.

Response: We have removed or shortened the methodology description for Figure 2. We have also provided full text for all abbreviations in the legend.

  1. Lines 154-159. The description of the methodology should be moved to the Materials and Methods chapter. In the Figure caption, it is desirable to add a transcript of all abbreviations and brief information necessary to understand the figure.

Response: We have removed or shortened the methodology description for Figure 3. We have also provided full text for all abbreviations in the legend.

  1. Lines 175-184. The description of the methodology should be moved to the Materials and Methods chapter. In the Figure caption, it is desirable to add a transcript of all abbreviations and brief information necessary to understand the figure.

Response: We have removed or shortened the methodology description for Figure 4. We have also provided full text for all abbreviations in the legend.

  1. Lines 197-203. The description of the methodology should be moved to the Materials and Methods chapter. In the Figure caption, it is desirable to add a transcript of all abbreviations and brief information necessary to understand the figure.

Response: We have removed or shortened the methodology description for Figure 5. We have also provided full text for all abbreviations in the legend.

  1. Lines 216-230. The description of the methodology should be moved to the Materials and Methods chapter. In the Figure caption, it is desirable to add a transcript of all abbreviations and brief information necessary to understand the figure.

Response: We have removed or shortened the methodology description for Figure 6. We have also provided full text for all abbreviations in the legend.

  1. Lines 267-272. The description of the methodology should be moved to the Materials and Methods chapter. In the Figure caption, it is desirable to add a transcript of all abbreviations and brief information necessary to understand the figure.

Response: We have removed or shortened the methodology description for Figure 7. We have also provided full text for all abbreviations in the legend.

  1. Lines 206-208. RPM, according to your data, are not cytotoxic, which contradicts the researchers' data, which is written at the beginning (lines 82-84). There is a need to explain what the differences in the results are related to. If your data are given for specific RPMs, it would be desirable to explain how they differ favorably from those RPMs that are described as cytotoxic.

Response: Our apologies for causing any confusion. In Lines 82-84 in the original submission, we were discussing the cytotoxicity of antimicrobial peptides (AMPs) in general. In contrast, we have shown that RPMs are not cytotoxic to airway epithelial cells and Caco2 cells in vitro (ACS Infect Dis 2021, 7 (3), 672-680; Microb Biotechnol 2018, 11 (6), 1027-1036), and low hemolysis (J Am Chem Soc 2013, 135 (32), 11748-51). Furthermore, maximum tolerated dose analysis of RPMs show that these peptide drugs do not have adverse effect on mouse weight, blood biochemistry and hematology (complete blood count with differentials), and histopathology of vital organs (Reference 46).

  1. In general, given the random nature of the assembly of peptides, there are two legitimate questions. Are the results reproducible, given the large number of random synthesis options? How possible is it that when analyzing the cytotoxicity of a mixture of peptides, a toxic peptide will be missed, which may be accidentally synthesized later?

Response: We have been working on RPMs in the last 8 years. During this time period, several students have synthesized them and we were able to observe great antimicrobial activity. These peptide mixtures are random but compose of two amino acids, so the number of the sequences is high but limited. We have characterized the RPMs after the synthesis using mass spectrometry and amino acid analysis. We have demonstrated the reproducibility in our earlier works (J Am Chem Soc 2013, 135 (32), 11748-51; and Chem Commun (Camb) 2020 56 (80) 12053-12056, in the Supporting Information sections) and have provided a comparison between several RPMs batches in term of chemical analysis and in vitro antimicrobial activity. In all the works we have published previously, we explored the antimicrobial activity after the RPMs synthesis and confirmed by chemical analysis.

In term of cytotoxicity, RPMs were extensively studied for their toxicity towards red blood cells, human cell lines, and in mouse model without any cytotoxic effect (also see response to Query #14 below).

  1. Is there a potential threat that foreign peptides that have entered the body will form into dangerous structures like prions?

Response: It is unlikely that RPMs will form dangerous structures. We have previously examined the in vitro cytotoxicity and in vivo toxicity in mice and found no evidence of harmful effects (see ACS Infect Dis 2021, 7 (3), 672-680).

  1. It is desirable to supplement the Introduction chapter with new data on research and development in the search for new methods to counteract bacterial resistance to antibiotics, for example, https://doi.org/10.3390/mi12121480 , https://doi.org/10.1080/17460441.2022.1985459 , https://doi.org/10.1186/s13662-021-03423-8.

Response: We have incorporated a brief description of new approaches, including metallic nanoparticles, pharmacophore analysis and computer-based modeling (Lines 89-95).